# Targeting the SREBP-1/Hsa-Mir-497/SCAP/FASN Oncometabolic Axis Inhibits the Cancer Stem-like and Chemoresistant Phenotype of Non-Small Cell Lung Carcinoma Cells

**DOI:** 10.3390/ijms23137283

**Published:** 2022-06-30

**Authors:** Tung-Yu Tiong, Pei-Wei Weng, Chun-Hua Wang, Syahru Agung Setiawan, Vijesh Kumar Yadav, Narpati Wesa Pikatan, Iat-Hang Fong, Chi-Tai Yeh, Chia-Hung Hsu, Kuang-Tai Kuo

**Affiliations:** 1Division of Thoracic Surgery, Department of Surgery, School of Medicine, College of Medicine, Taipei Medical University, Taipei 11031, Taiwan; 09262@s.tmu.edu.tw; 2Division of Thoracic Surgery, Department of Surgery, Taipei Medical University—Shuang Ho Hospital, New Taipei City 23561, Taiwan; 3Department of Orthopaedics, School of Medicine, College of Medicine, Taipei Medical University, Taipei 11031, Taiwan; wengpw@tmu.edu.tw; 4Department of Orthopaedics, Taipei Medical University—Shuang Ho Hospital, New Taipei City 23561, Taiwan; 5Graduate Institute of Biomedical Materials and Tissue Engineering, College of Biomedical Engineering, Taipei Medical University, Taipei 11031, Taiwan; 6School of Medicine, Buddhist Tzu Chi University, Hualien 970, Taiwan; 10205@s.tmu.edu.tw; 7Department of Dermatology, Taipei Tzu Chi Hospital, Buddhist Tzu Chi Medical Foundation, New Taipei City 231, Taiwan; 8International Ph.D. Program in Medicine, College of Medicine, Taipei Medical University, Taipei City 11031, Taiwan; setiawan.syahru@gmail.com; 9Department of Medical Research & Education, Taipei Medical University—Shuang Ho Hospital, New Taipei City 23561, Taiwan; vijeshp2@gmail.com (V.K.Y.); impossiblewasnothing@hotmail.com (I.-H.F.); ctyeh@s.tmu.edu.tw (C.-T.Y.); 10Division of Urology, Department of Surgery, Faculty of Medicine, Universitas Gadjah Mada, Yogyakarta 55281, Indonesia; narpatisesa@gmail.com; 11Department of Medical Laboratory Science and Biotechnology, Yuanpei University of Medical Technology, Hsinchu 300, Taiwan; 12Department of Emergency Medicine, Taipei Medical University—Shuang Ho Hospital, New Taipei City 23561, Taiwan; 13Graduate Institute of Injury Prevention and Control, College of Public Health, Taipei Medical University, Taipei City 11031, Taiwan; 14Department of Emergency Medicine, School of Medicine, Taipei Medical University, Taipei 110, Taiwan

**Keywords:** SREBP-1, SCAP, NSCLC, metabolism, chemoresistance

## Abstract

Background: Lung cancer remains a leading cause of cancer-related death, with an annual global mortality rate of 18.4%. Despite advances in diagnostic and therapeutic technologies, non–small cell lung carcinoma (NSCLC) continues to be characterized by a poor prognosis. This may be associated with the enrichment of cancer stem cells (CSCs) and the development of chemoresistance—a double-edged challenge that continues to impede the improvement of long-term outcomes. Metabolic reprogramming is a new hallmark of cancer. Sterol regulatory element-binding proteins (SREBPs) play crucial regulatory roles in the synthesis and uptake of cholesterol, fatty acids, and phospholipids. Recent evidence has demonstrated that SREBP-1 is upregulated in several cancer types. However, its role in lung cancer remains unclear. Objective: This study investigated the role of SREBP-1 in NSCLC biology, progression, and therapeutic response and explored the therapeutic exploitability of SREBP-1 and SREBP-1-dependent oncometabolic signaling and miRNA epigenetic regulation. Methods: We analyzed SREBP-1 levels and biological functions in clinical samples and the human NSCLC cell lines H441 and A549 through shRNA-based knock down of SREBP function, cisplatin-resistant clone generation, immunohistochemical staining of clinical samples, and cell viability, sphere-formation, Western blot, and quantitative PCR assays. We conducted in-silico analysis of miRNA expression in NSCLC samples by using the Gene Expression Omnibus (GSE102286) database. Results: We demonstrated that SREBP-1 and SCAP are highly expressed in NSCLC and are positively correlated with the aggressive phenotypes of NSCLC cells. In addition, downregulation of the expression of tumor-suppressing hsa-miR-497-5p, which predictively targets SREBP-1, was observed. We also demonstrated that SREBP-1/SCAP/FASN lipogenic signaling plays a key role in CSCs-like and chemoresistant NSCLC phenotypes, especially because the fatostatin or shRNA targeting of SREBP-1 significantly suppressed the viability, cisplatin resistance, and cancer stemness of NSCLC cells and because treatment induced the expression of hsa-miR-497. Conclusion: Targeting the SREBP-1/hsa-miR-497 signaling axis is a potentially effective anticancer therapeutic strategy for NSCLC.

## 1. Introduction

Lung cancer is the most common cancer and the leading cause of cancer-related death worldwide [1]. Approximately 80% of lung cancer diagnoses, especially adenocarcinoma, are pathologically classified as non–small cell lung cancer (NSCLC) [1]. For patients with advanced-stage disease, chemotherapy remains the preferred treatment option [2]. In the last decade, resistance to chemotherapy and epidermal growth factor receptor tyrosine kinase inhibitors (EGFR-TKI) has emerged as a clinical challenge [3]. The development of chemoresistance continues to impede the improvement of the long-term outcomes of patients with advanced lung adenocarcinoma [4,5]. Therefore, elucidation of the mechanisms underlying the chemoresistance of lung adenocarcinoma cells is necessary to facilitate the identification of new therapeutic targets, development of effective anticancer therapies, mitigation of therapy failure, and improvement of the prognosis of patients with lung adenocarcinoma.

The cancer stem cells (CSCs) hypothesis has attracted much attention over the last 3 decades. CSCs are inherently self-renewing, stress- and drug-resistant, and characterized by enhanced migration [6,7]. These features may promote chemoresistance, metastasis, and tumor recurrence. Lung CSCs (LCSCs) were first identified as CD133+/Oct4+/Nanog+ cells isolated from established NSCLC cell lines [8]. LCSCs have been reported to actively efflux Hoechst 33,342 dye, exhibit high aldehyde dehydrogenase (ALDH) activity, and are designated as side population cells in flow cytometric assays [9,10]. In addition, CSCs, which are a subset of intratumoral cells that promote tumor growth and metastasis, express high levels of ATP binding cassette subfamily G member 2 protein (ABCG2), a multidrug transporter, and resist TKI treatment by modulating intracellular TKI concentrations [11]. Researchers have suggested that LCSCs play a crucial role in chemoresistance, metastasis, and tumor progression. Therefore, investigating the molecular mechanisms of LCSC development may be beneficial for the development of effective therapeutic strategies for lung cancer.

Metabolic reprogramming is a new hallmark of cancer [12]. Increasing evidence demonstrates that alterations in lipid metabolism are often present in cancer cells and promote tumor growth [13]. Lipids serve as basic structures in the plasma membrane and the membranes of most cellular organelles. In addition, lipids serve as energy resources and function as key signaling molecules, regulating various cellular functions, including migration and differentiation as well as cell division and other steps of the cell cycle [14,15]. A markedly increased lipid requirement is associated with the rapid growth and proliferation of tumor cells [16,17]. Lipid synthesis and uptake are highly elevated in various cancers [18,19]. These processes are regulated by sterol regulatory element-binding proteins (SREBPs), which are endoplasmic reticulum (ER)-bound transcription factors that control the expression of genes important for the synthesis and uptake of cholesterol, fatty acids, and phospholipids [20]. Researchers have identified two mammalian SREBP genes: SREBF1 and SREBF2. SREBP-1a and SREBP-1c, which are encoded by SREBF1 and have different N-termini (of approximately 20 amino acids) due to the initiation of transcription at different transcriptional start sites, mainly regulate the expression of genes required for fatty acid synthesis [21]. SREBP-2 is encoded by SREBF2 and is responsible for the synthesis of cholesterol [22]. A recent study suggested that the nuclear form of SREBP-1 is highly upregulated in several cancer types, including glioblastoma (GBM), cervical cancer, and hepatocellular carcinoma (HCC) [23]. The transcriptional activation of SREBPs requires binding to SREBP cleavage-activating proteins (SCAPs) for the translocation of inactive precursors from the ER to the Golgi apparatus, where they undergo cleavage and subsequent nucleus translocation of their NH2-terminal forms [15]. Recent reports that SREBPs are markedly upregulated in human cancers provide some mechanistic insights into the probable link between altered lipid metabolism and carcinogenesis [24]. Impaired expression or activity of SREBPs and their downstream target genes has been implicated in various cancer types. For instance, SREBPs-mediated upregulation of fatty acid synthase (FASN) and cholesterol favors the Warburg effect, which is the characteristic modification of cellular metabolism in malignant cells [25,26]. Increasing evidence that the pharmacological or genetic inhibition of SCAPs or SREBPs significantly suppresses tumor growth in various cancer models [25,26], informs our position that SCAPs/SREBPs may be promising metabolism-related molecular targets for the treatment of patients with lung cancer.

The integrative role of microRNAs (miRNAs)—small (22-nucleotide) noncoding RNAs that regulate genes by binding to the 3′-untranslated region (UTR) of target mRNA [27]—in lipid metabolism cannot be ignored. Recent studies have demonstrated the key role of miRNAs in regulating lipid metabolism and many important miRNAs, such as miR-33, miR-122, miR-370, and miR-27 [27,28]. Importantly, miR-122 plays a role in the regulation of fatty acid and triglyceride biosynthesis, such as fatty acid synthesis (FAS), and in controlling the levels of acetyl-CoA carboxylase 1 (ACC-1) and sterol regulatory element-binding protein 1c (SREBP-1c) and the expression of many genes that regulate fatty acid β-oxidation [29]. Similarly, other miRNAs regulate homeostasis by targeting lipids associated with metabolism-related genes.

The present study investigated the molecular mechanism underlying the SREBP-mediated stemness and chemoresistance of lung cancer cells. In addition to identifying candidate metabolites involved in SREBP-mediated lung cancer stemness and chemoresistance, we determined that the ensuing CSCs-like phenotype is induced by SREBP/SCAP-induced lipid metabolism reprogramming, and that this reprogramming therefore plays a key role in the development of chemoresistance by lung cancer cells. Additionally, our in-silico analysis of NSCLC miRNA conducted using the public Gene Expression Omnibus (GEO; GSE102286) database demonstrated that the expression of hsa-miR-497, which predictively targets the expression of SREBP-1, was significantly inhibited in our tumor samples. Finally, we developed therapeutic strategies involving the inhibition of SREBP-1 and SCAP signaling and the induction of hsa-miR-497 expression to inhibit lung cancer chemoresistance and progression.

## 2. Results

### 2.1. SREBP-1 and SCAP Levels Are Elevated in Patients with NSCLC

To understand the biological significance of SCAP in the development and progression of lung cancer, especially in patients with NSCLC, we evaluated the levels of SCAP and its ligand SREBP-1 in lung cancer specimens (*n* = 98) through IHC staining (Figure 1A,B). The results indicated that the levels of expression of both SREBP-1 and SCAP in the NSCLC samples were higher than those in the non-tumor samples, and that these elevated levels were stage-dependent (Figure 1C,D). Similarly, our confirmation through western blotting also observed that SREBP-1 and SCAP were highly expressed in tumor site than non-tumor adjacent site of NSCLC samples (Appendix A). Interestingly, predominant nuclear localization was observed in lung tumor samples, which stipulated an intensive nuclear translocation of this transcription factor in lung malignancy (Figure 1E). To compare our finding with other report, public microarray profiling dataset (GSE18842) was utilized that previously described the expression profiling of NSCLC [30]. Re-analysis of this dataset resulted that SREBP-1 level was significantly elevated in adenocarcinoma (Adeno-Ca group) and squamous cell lung carcinoma (SCC group) samples relative to those in the normal lung tissue samples; however mRNA level of SCAP did not have significant different among those groups (Figure 1F). Correlation analysis revealed that SREBP-1 levels were significantly correlated with SCAP levels in the GSE18842 samples (*p* < 0.001; *R* = 0.44; Figure 1G).

### 2.2. SREBP-1 and SCAP Expression Is Positively Associated with the Aggressive Phenotype of NSCLC Cells

To gain clinicopathological insights into the role of *SREBP-1* expression in NSCLC, after dividing the samples into two groups (according to whether they had high or low SREBP-1 levels, using the median level as the cut-off), we investigated whether *SREBP-1* expression was associated with clinicopathological characteristics in our cohort (*n* = 98). The association between SREBP-1 expression and clinical features among patient with NSCLC was described in Table 1. The gender of patient, age, and history of smoking did not associate significantly with expression of SREBP-1 among NSCLC tissue (*p* > 0.05). Interestingly, higher SREBP-1 expression was significantly associated to poor tumor differentiation (OR 2.39; CI 95% 1.05–5.43; *p* = 0.035) and advanced TNM stage (OR 3.63; CI 95% 1.56–8.41; *p* = 0.002). In addition, we identified significant association of higher SCAP expression and elevation of SREBP-1 level in NSCLC tissue (OR 2.34; CI 95% 1.02–5.36; *p* < 0.042).

### 2.3. SREBP-1-Dependent Lipogenic Signaling Plays a Key Role in the CSCs-Like and Chemoresistant Phenotypes of NSCLC Cells

After demonstrating that SREBP-1 was overexpressed in NSCLC cells, we investigated the likelihood that SREBP-1 overexpression is implicated in the CSCs-like and chemoresistant phenotypes of NSCLC cells. Using the wild-type and cisplatin-resistant (R) human NSCLC cell lines H441 and A549, we demonstrated that although the H441 and A549 cells were sensitive to cisplatin treatment, with IC_50_ concentrations of 58 µM and 47 µM, respectively, H441R and A549R cells were less responsive to the anticancer cytotoxicity of cisplatin, with a concentration of 120 µM producing a cytotoxicity rate of approximately 50% in both cell lines (Figure 2A). Furthermore, we demonstrated that the levels of CD133 were significantly higher in H441R and A549R cells than in their wild-type counterparts (13.36-fold higher, *p* < 0.05 and 14.15-fold higher, *p* < 0.05, respectively; Figure 2B), with enhanced capacity for tumorsphere formation (H441R vs. H441: 2.84-fold higher, *p* < 0.05; A549R vs. A549: 4.25-fold higher, *p* < 0.05; Figure 2C). In addition, H441R and A549R cells formed more colonies than did their wild-type counterparts (1.65-fold more, *p* < 0.05 and 2.03-fold more, *p* < 0.05, respectively; Figure 2D). The CSCs-like phenotype was strongly associated with the significant upregulation of CD133, SREBP-1, FASN, SCAP, HMGCR, HMGCS, and ABCG2 mRNA expression levels (Figure 2E). These results are, at least in part, indicative of the crucial role of SCAP-dependent lipogenic signaling in the LCSCs and chemoresistant phenotypes of NSCLC cells.

### 2.4. Fatostatin-Induced Targeting of SREBP-1 Signaling Significantly Inhibits Cisplatin Resistance and Cancer Stemness in NSCLC Cells

After demonstrating the key role of SCAP-dependent lipogenic signaling in LCSCs and chemoresistant phenotypes of NSCLC cells, we examined the effect of targeting SREBP, an essential component of the SCAP-dependent lipogenic signaling pathway, by using fatostatin, an SREBP inhibitor. Pretreatment of H441R and A549R cells with 5 µM fatostatin for 24 h enhanced their sensitivity to cisplatin, with IC_50_ concentrations of 59 µM and 73 µM, respectively (Figure 3A). Furthermore, the results of western blot analysis showed that treatment with 5 µM Fatostatin reduced expression of cleavage SREBP-1 in the nuclear compartment, along with reduction of SCAP, INSIG1, FASN and HMGCS-1 proteins. On the other hand, the protein expression of cytoplasmic SREBP-1 as precursor form did not change as depicted in the western blot result (Figure 3B). Consistently, Fatostatin treatment also reduced mRNA expression of the downstream of SREBP-1/SCAP axis such as INSIG1, FASN, and HMGCS-1 (Figure 3C). Consequently, treatment with 5 to 10 µM fatostatin significantly reduced the percentage of CD133+ cells (Figure 3D) and the capability of the treated cells to form tumorspheres, as indicated by the presence of fewer and smaller spheres than were observed for the untreated control cells (Figure 3E).

### 2.5. Silencing SREBP-1 in Cisplatin-Resistant NSCLC Cells Reduces the Viability and Stemness of NSCLC Cells

We examined the effects of shRNA-induced silencing of *SREBP-1* in H441R and A549R cells (Figure 4A). Compared with their wild-type counterparts, shSREBP-1-transfected H441R and A549R cells exhibited significantly higher sensitivity to cisplatin in a dose-dependent manner (Figure 4A, lower panel). In addition, relative to the vector control (VC) cells, shSREBP-1-transfected H441R and A549R cells formed significantly smaller and fewer tumorspheres (2.8-fold, *p* < 0.05 and 3.0-fold, *p* < 0.05, respectively; Figure 4B). Consistent with earlier results, transfection with shSREBP-1 significantly reduced the percentage of CD133+ H441R and A549R cells (by 31%, *p* < 0.05 and 47%, *p* < 0.05, respectively; Figure 4C). Moreover, shSREBP-1 markedly suppressed the expression of SREBP-1, SCAP, FASN, and SREBP2 in H441R and A549R cells (Figure 4D). To confirm the finding, we also compared the perturbation following shRNA-mediated knockdown of SREBP-1 between unsorted parental cell lines and resistant clone that also resulted in an attenuated CD133 activity, reduced generation of tumor sphere, and suppression of colony formation in A549 NSCLC cell line (Appendix A). Therefore, these results might indicate to some extent that SREBP-1 determined metabolic axis to provoke stemness phenotype of NSCLC. In addition, relative to their VC counterparts, shSREBP-1-transfected H441R and A549R cells exhibited profound inhibition of stearoyl-CoA desaturase 1 (SCD1) expression with the concomitant upregulation of FASN, poly(ADP-ribose) polymerase (PARP), and phosphorylated eukaryotic initiation factor 2 (peIF2) mRNA expression (Figure 4E), suggesting enhanced lipotoxicity, accumulation of reactive oxygen species (ROS), and ER stress.

### 2.6. Increased SREBP-1/SCAP/FASN Signaling Is Implicated in the Reduced Sensitivity of NSCLC Cells to Cisplatin

Finally, we validated our in vitro findings by using our own NSCLC cohort, which consisted of 20 pairs of cisplatin-sensitive and cisplatin-resistant clinical samples. Consistent with our aforementioned findings, according to the results of IHC staining, the cisplatin-resistant samples exhibited significantly higher levels of SREBP-1 (1.5-fold, *p* < 0.001), SCAP (1.86-fold higher, *p* < 0.001), and FASN (1.76-fold, *p* < 0.001) expression than did the cisplatin-sensitive samples (Figure 5A). The qRT-PCR analysis supported the IHC staining results at the mRNA level (Figure 5B).

### 2.7. miR-497-5p Inhibits SREBP-1/SCAP Signaling by Binding to the 3′UTR of Target mRNA and Induces Cisplatin Sensitivity in NSCLC Cells

We explored the role of miRNAs in NSCLC by constructing a heat map (Figure 6A). In our miRNA analysis, the expression of 66 miRNAs differed significantly between the healthy samples and tumor samples, with 23 and 43 of the differentially expressed (DE) miRNAs being significantly upregulated and downregulated, respectively, in the tumor samples, as indicated in the volcano plot in Figure 6B. We explored the predicted miRNA targets of SREBP-1 and SCAP by using the ENCORE database (https://starbase.sysu.edu.cn/index.php) and performed intersection analyses of DE miRNAs (Figure 6C). The results of the intersection analysis indicate that hsa-miR-497-5p targets both SREBP-1 and SCAP binding at the 3′ UTR. In addition, we performed a qRT-PCR (Figure 6D) and Western blot (Figure 6E) analysis of cisplatin-resistant lung cancer cells transfected with a mimic and control, and the results indicated that hsa-miR-497-5p expression was inversely associated with SREBP-1 expression in both types of cells.

## 3. Discussion

SREBP-1 plays a key role in NSCLC metabolism and is involved in the maintenance of the stemness and chemoresistance of NSCLC cells. Using the clinical samples from our NSCLC cohort and GSE18842 dataset, we demonstrated that aberrant expression of SREBP-1 and SCAP was associated with disease progression and a poor prognosis in lung cancer, thus indicating their potential clinical utility as reliable diagnostic or prognostic biomarkers and as potential molecular targets for the treatment of lung adenocarcinoma. We demonstrated that prolonged exposure to cisplatin desensitized NSCLC cells to cisplatin, leading to the development of cisplatin-resistant NSCLC cells with CSC-like phenotypes that are characterized by enhanced SREBP-1-mediated lipogenesis. Furthermore, we demonstrated that the inhibition of SREBP-1 by fatostatin or through shRNA- or miRNA (hsa-miR-497-5p)-mediated regulation increased the sensitivity of cisplatin-resistant cells to cisplatin and suppressed the cancer stemness of lung cancer cells. Our results may serve as a reference for the design of future clinical trials of therapeutic modalities that target the SREBP-1/hsa-miR-497-5p/SCAP/FASN oncometabolic signaling axis to overcome chemoresistance in patients with lung cancer.

Over the last decade, metabolic reprogramming has emerged as a key hallmark of cancer and has been determined to be modulated by SREBP-1 [31]. The present study demonstrated that SREBP-1 and SCAP levels were positively correlated with disease progression and a poor prognosis in lung cancer. SREBP-1 expression was significantly correlated with tumor differentiation and TNM stage but was not associated with sex, age, or smoking history. This finding is consistent with recent reports that the SREBP-1 escort protein SCAP was overexpressed in bladder cancer tissues, and that this overexpression was positively associated with tumor invasion, lymph node metastasis, TNM stage, and histological grade in patients with bladder cancer. The same study identified no direct association between SREBP-1 expression and age or sex [32]. This study demonstrated that SREBP-1 was highly upregulated in poorly differentiated tumor and advanced stage of NSCLC. In line with the previous report, this finding highlights aberrant activation of SREBP-1/SCAP axis to amplify cancer stem cells that are lacking differentiation and depend on lipogenesis, as this axis is also the master regulator of lipid metabolism [32,33]. Poorly differentiated cancers generally increased proliferation rate, thus, frequently found in advanced stage with high incidence of metastasis and are highly invasive.

In the present study, we observed elevated SREBP-1 expression in the cisplatin-resistant tumor samples and discovered that silencing SREBP-1 enhanced the cisplatin sensitivity of NSCLC cells, indicating that SREBP-1 plays a key role in chemoresistance. This finding is in line with reports that SREBP-1 inhibition suppressed cellular glucose metabolism, reduced glycolytic activity, and inhibited the metastatic potential of HCC cells in addition to promoting the anticancer effects of sorafenib on HCC cells and xenograft tumors [34]. This is further corroborated by our finding that SREBP-1, SCAP, and FASN expression in the cisplatin-resistant samples was higher than that in their cisplatin-sensitive counterparts. Wang et al. discovered that blocking fatty acid β-oxidation and leptin could resensitize breast cancer cells to chemotherapy and could inhibit the development of breast CSCs in mouse breast tumors in vivo [35]. Consistent with our findings, another study reported that SREBP-1 was overexpressed in chemoresistant colorectal carcinoma (CRC) samples, and that SREBP-1 overexpression downregulated the expression of caspase 7, decreased CRC cell sensitivity to gemcitabine, and was positively correlated with a poor prognosis. The same study demonstrated that targeting SREBP-1 enhanced the sensitivity of CRC cells to gemcitabine, and that low SREBP-1 expression was correlated with elevated expression of caspase-7 in CRC samples [36]. Similarly, Chen et al. conducted a study using a bone-cancer dual-targeting biomimetic nano-delivery system and demonstrated that siRNA targeting of SREBP-1 enhanced the therapeutic effect of docetaxel against bone-metastatic castration-resistant prostate cancer [37].

Recent advances in cancer metabolism research have demonstrated the importance of cancer metabolism to the survival of cancer cells, particularly to CSCs, which constitute a subset of the cancer cell population and contribute to cancer cell proliferation and chemoresistance and treatment failure. Over the last 2 decades, the distinct pattern of metabolism in CSCs has been more thoroughly documented. A recent study demonstrated that the enhanced oncogenicity of glioblastoma cells is mainly dependent on glycolysis, and that CSCs-like cells are often resistant to conventional chemotherapy but sensitive to glycolytic inhibition [38]. Similar studies have demonstrated that ovarian CSCs-like spheroid cells mainly rely on anaerobic glycolysis [39,40]. CSCs exhibit elevated fatty acid metabolic activity, which may be correlated with cancer malignancy. In ovarian cancer, fatty acid β-oxidation (FAO) activity in CSCs is higher than that in non-CSCs [41]. This is consistent with our findings that enhanced SREBP-1/SCAP/FASN signaling is implicated in the reduced sensitivity of NSCLC cells to cisplatin therapy, and that the viability and stemness of these resistant NSCLC cells can be modulated through shRNA-mediated silencing of SREBP-1 or the introduction of exogenous hsa-miR-497-5p into cisplatin-resistant NSCLC cells. A recent study reported that SREBP-1 knockdown decreased fatty acids levels by decreasing the expression of SREBP-targeted genes required for lipid biosynthesis in CRC cells [24]. SREBP-1 knockdown was also associated with altered cellular metabolism, including decreased mitochondrial respiration, glycolysis, and FAO as well as a significant decrease in the cell proliferation rate and the ability of CRC cells to form spheres in a suspension culture [24]. In addition, a study reported that reprogramming of CSCs metabolism by restoring oxidative phosphorylation or FAO inhibition is a potentially effective anticancer therapeutic strategy, as indicated by the authors’ findings that restoring oxidative phosphorylation or inhibiting FAO in HCC could sensitize liver CSCs to sorafenib treatment [34]. A study of patients with breast cancer reported that elevated FAO activity contributed to self-renewal and chemoresistance in breast cancer cells [35]. Thus, the aforementioned findings indicate that SREBP-1 knockdown reduces the ability of NSCLC cells to form spheres and is associated with decreased levels of ALDH1 and SREBP-1-regulated FAS biomarkers, elevated ROS levels, and enhanced phosphorylation of eIF2.

## 4. Materials and Methods

### 4.1. Cell Culture and Transfection

The human NSCLC cell lines H441 and A549 were obtained from the American Type Culture Collection (Rockville, MD, USA). H441 cells were grown in RPMI 1640 medium supplemented with 10% fetal bovine serum and 2-mM L-glutamine. A549 cells were grown in DMEM/F12 medium supplemented with 10% fetal bovine serum and 4-mM L-glutamine. All the cell culture reagents were from Hyclone (Logan, UT, USA). The cells were subcultured to 90% confluence, and the cell media were replaced every 48 h. For subsequent assays, including flow cytometry, cells were treated with Accutase (Stemcell Technologies, Vancouver, BC, Canada). All the cells were cultured at 37 °C in a 5% CO_2_ atmosphere incubator. The H441 and A549 NSCLC cell lines were transfected with short-hairpin RNAs (shRNAs) targeting SREBP-1, specifically at the following sequences: ATCGCTTGCTTCATCGATATT (clone 1), GTGCCTGTTTACCGAACTAAT (clone 2), or GCACCAAATTAGAGAGTCT (clone 3) within DNA sequence of SREBP-1 or a vector (pLKO_TRC005) through Lipofectamine Plus (Invitrogen, Thermo Fisher Scientific, Waltham, MA, USA) according to the manufacturer’s protocol. The shRNAs were obtained from the National RNAi Core Facility at Academia Sinica, Taipei City, Taiwan. Adherent cells were treated with 0.5 mL of the virus, followed by overnight incubation at 37 °C in a 5% CO_2_ atmosphere incubator. On the following day, the viral medium was replaced with fresh medium, and the culture was treated with puromycin (1 µg/mL) to select a population of stably transfected cells.

### 4.2. Establishment of Cisplatin-Resistant Lung Cancer Cell Lines

We obtained cisplatin-resistant NSCLC cells (H441 and A549) by applying slightly modified versions of the protocols proposed by Barr et al. (2020). [42] We conducted initial dose–response studies, in which the cells were exposed to cisplatin (0.1–100 µM) for 72 h to obtain the IC_50_ values (the drug concentrations that resulted in the death of 50% of the cultured cells). Thereafter, NSCLC cells were treated with cisplatin (IC_50_) for 72 h. The medium was changed, and the cells were allowed to recover for an additional 72 h. This treatment continued for approximately 6 months, after which the IC_50_ concentrations for each resistant cell line were reevaluated. The cells were then exposed continuously to cisplatin at the new IC_50_ concentrations for an additional 6 months.

### 4.3. Tumorsphere Formation Assay

For two weeks, the NSCLC cell lines (H441 and A549) were grown as spheres in a 10-cm ultra-low adhesion culture dish (Corning, Glendale, AZ, USA) containing DMEM/F-12 with N2 supplement (Invitrogen, Thermo Fisher Scientific, Waltham, MA, USA), 20 ng/mL EGF, and 20 ng/mL basic fibroblast growth factor (FGF; PeproTech, Rehovot, Israel), also known as stem cell media. The ratio of sphere number to plated cell number was used to calculate the tumor sphere formation efficiency.

### 4.4. Activity of CD133 by Flow Cytometry

H441 and A549 cell lines were washed with 1 × PBS and harvested prior to antibody incubation. Flow cytometry technique was used to examine CD133 expression on single cells detached from plate. For 45 min, one million trypsinized cells were treated with an anti-CD133 antibody (Cell Signaling Technology Inc., Beverly, MA, USA) or an isotype control IgG (Upstate Biotechnology, Lake Placid, NY, USA). Following washing, the cells were incubated for 30 min with an Alexa488/Alexa594-conjugated secondary antibody (Invitrogen, Thermo Fisher Scientific, Waltham, MA, USA) before being analyzed with a BD FACScaliber flow cytometer (BD Biosciences, Franklin Lakes, NJ, USA). Cell Quest Software was used to examine the fluorescent intensity (BD Biosciences, Franklin Lakes, NJ, USA).

### 4.5. Immunohistochemistry

This study was conducted in a cohort of patients with lung cancer who underwent resection at Taipei Medical University Shuang Ho Hospital in Taipei, Taiwan, between January 2010 and December 2017. A predesigned data collection format was used to review the patients’ medical records for evaluation of their clinicopathological characteristics and survival outcomes. The study of patient’s samples was approved by the Taipei Medical University-Shuang Ho Hospital (Approval no.: JIRB N201801066) and complied with the recommendations of the Declaration of Helsinki for Biomedical Research. Clinical samples from the patients were fixed in 10% formalin, embedded in paraffin, deparaffinized, and rehydrated. For immunohistochemical (IHC) staining, the rehydrated sections were subjected to antigen retrieval, and their endogenous peroxidase activity was blocked for 30 min in 1% hydrogen peroxide/phosphate-buffered saline (H_2_O_2_/PBS) solution. After blocking, the slides were exposed to SCAP or SREBP antibodies at 1:200 dilution overnight at 4 °C, washed, and incubated in biotinylated link universal antiserum for 1 h at room temperature. The slides were then rinsed, and staining was developed using chromogen, 3,3- diaminobenzidine hydrochloride. Finally, the sections were rinsed with double-distilled water and counterstained with hematoxylin. The slides were examined under a light microscope in five random fields of view. Evaluation and quantification of the SCAP and SREBP levels were performed manually by two independent investigators. The staining intensity was graded as 1, 2, or 3, corresponding to absent or weak, moderate, and strong staining, respectively. IHC staining was quantified according to the Quick score (*Q* score) formula: *Q* = *P* × *I*, where *P* is the percentage of stained tumor cells (0–100%), and *I* is the staining intensity (1–3), producing a maximum *Q* score of 300. For survival analysis, we used a *Q* score of 150 as the cut-off value to divide the patients into two groups.

### 4.6. MTT Cell Viability Assay

An MTT assay was performed on monolayer cultures following the protocol describes procedures. After 4 days of treatment with the indicated therapeutic agent, 20 µL of MTT solution was added into each well containing a monolayer or multicellular tumor spheroid (MCTS) culture, and the cultures were incubated for 4 h. The monolayer cultures were left untouched in the original plates, whereas the content of each well containing the MCTS culture was transferred to new flat-bottom 96-well plates and were then centrifuged at 1000× *g* for 5 min. Thereafter, 150 µL of medium was aspirated from each well containing a monolayer or MCTS culture. The plates were then blot-dried using paper towels, followed by the addition of 100 µL of dimethyl sulfoxide (DMSO). Finally, absorbance at 570 nm was recorded using a µQuant Enzyme-Linked Immunosorbent Assay (ELISA) Reader (Bio-tek Instruments, Winooski, VT, USA). The IC_50_ concentrations of the monolayer and MCTS cultures were determined according to the dose–response curves. The assay was performed in 12 replicates for each culture.

### 4.7. Preparation of Nuclear and Cytoplasmic Lysates

The NSCLC cell line-treated and -untreated cells (H441 and A549) were harvested and centrifuged prior to extraction of nuclear and cytoplasmic lysates. The cells were then lysed using NE-PER nuclear and cytoplasmic extraction reagents (Thermo Fisher Scientific, Carlsbad, CA, USA) according to the manufacturer’s instructions after being washed twice with ice-cold 1 × PBS. The bicinchoninic acid (BCA) protein assay (Pierce, Rockford, IL, USA) was used to assess the protein contents in all of the extracts, using bovine serum albumin as the reference. Each extract was kept at −70 °C until it was used.

### 4.8. Western Blot Analysis

Cell lysates were extracted using Pierce Radioimmunoprecipitation Assay (RIPA) Buffer (Cat. #89900, Thermo Fisher Scientific, Carlsbad, CA, USA) and a 1% Halt 158 Protease Inhibitor Cocktail Kit (Cat. #78410, Thermo Fisher Scientific, Waltham, MA, USA). The cell extracts were loaded onto Mini159 Protean TGX 4–15% gels (Bio-Rad, Hercules, CA, USA) and were transferred using the Trans-Blot Turbo Blotting 160 System (Bio-Rad, Hercules, CA, USA). The cells were incubated with primary antibodies against SREBP-1 (Cat. # ab3259, Abcam, Cambridge, MA, USA), SCAP (Cat. # ab190103, Abcam, Cambridge, MA, USA), INSIG1 (Cat. # ab70784, Abcam, Cambridge, MA, USA), FASN (Cat. # ab128870, Abcam), HMGCS1 (Cat. # ab155787, Abcam, Cambridge, MA, USA), LAMIN-B1 (Cat. # ab16048, Abcam, Cambridge, MA, USA), Vimentin (Cat. # 5741, Cell Signaling, Beverly, MA, USA), N-Cadherin (Cat. # 13116, Cell Signaling, Beverly, MA, USA), KLF4 (Cat. # ab129473, Abcam, Cambridge, MA, USA), NANOG (Cat. # ab109250, Abcam, Cambridge, MA, USA), OCT4 (Cat. # ab200834, Abcam, Cambridge, MA, USA), CD133 (Cat. # ab222782, Abcam, Cambridge, MA, USA), FASN (Cat. # ab128870, Abcam, Cambridge, MA, USA), β-actin (Cat. # 66009-1-Ig, Proteintech, Rosemont, IL, USA), and GAPDH (Cat. # ab8245, Abcam, Cambridge, MA, USA) overnight at 4 °C. The primary antibodies are as enlisted in Appendix A. The cells were then incubated with secondary antibodies and the ECL substrate (Bio-Rad, Hercules, CA, USA), and the protein bands were visualized using a ChemiDoc 164 TM Touch 165 Imaging System (Bio-Rad, Hercules, CA, USA). The images were processed using Image Lab Software for PC, version 5.2.1 (Bio-Rad, Hercules, CA, USA).

### 4.9. Quantitative Real-Time Polymerase Chain Reaction

Total RNA was extracted using the Ambion TRIzol Reagent (Thermo Fisher Scientific, Waltham, MA, USA), and complementary DNA was synthesized from 1.0 μg of the total RNA by using the oligo dT primer and a reverse transcription system (Cat. #A3500, Promega, Madison, WI, USA) according to the manufacturer’s instructions. A quantitative real-time polymerase chain reaction (qRT-PCR) was performed using Light Cycler Faststart DNA Master SYBR Green I (Roche Diagnostics, Indianapolis, IN, USA) and a LightCycler TaqMan Master (Roche Diagnostics, Indianapolis, IN, USA) on LightCycler 2.0 II (Roche Molecular Systems, Branchburg, NJ, USA). Intergroup differences in target gene expression were determined using the 2^−ΔΔCt^ method.

### 4.10. Statistical Analysis

Data are represented as the mean ± standard deviation except where indicated otherwise. Unpaired Student’s t test was performed to compare mean values between the groups. One-way analysis of variance (ANOVA) was used to compare values among different groups. When ANOVA results were significant, post hoc testing of differences between groups was performed using the least significant difference test. Two-way ANOVA with repeated measures was employed to compare different groups under different timelines. Pearson’s linear correlation was used to determine the correlation strength between different parameters. Logistic regression was used to determine association of independent variables comprising of clinical characteristics of non-small cells lung cancers patients with an outcome variable such as the SREBP-1 and SCAP level group. Odd ratio (OR) was calculated to indicate positive or negative association of each predictor to occur outcome. A *p* value of <0.05 was considered statistically significant. All statistical analyses were performed using R studio (version 1.4.1717, Boston, MA, USA) and GraphPad Prism (version 8.02, San Diego, CA, USA).

## 5. Conclusions

In summary, as shown in schema abstract Figure 7, we herein report that the SREBP-1/hsa-miR-497-5p/SCAP/FASN oncometabolic signaling axis plays a major role in the stemness and chemoresistance of NSCLC cells. SREBP-1 and SCAP expression are strongly associated with disease progression and a poor prognosis in NSCLC, making them reliable biomarkers and potential molecular targets for the treatment of lung adenocarcinoma. This study has some limitations, including its small cohort size. The results obtained in this study should be tested in future studies with larger or multicenter cohorts of patients with more diverse clinicopathological characteristics and clinical outcomes to further evaluate the potential of SREBP-1 as a prognostic marker in NSCLC. The findings of this study provide additional preclinical evidence that targeting the SREBP-1/hsa-miR-497-5p SCAP/FASN oncometabolic signaling axis may be an effective therapeutic strategy for NSCLC. In addition, the present study provides a foundation for the design of future clinical trials evaluating the efficacy of targeting SREBP-1/hsa-miR-497-5p SCAP/FASN signaling for the treatment of chemoresistant lung cancer.

## Figures and Tables

**Figure 1 ijms-23-07283-f001:**
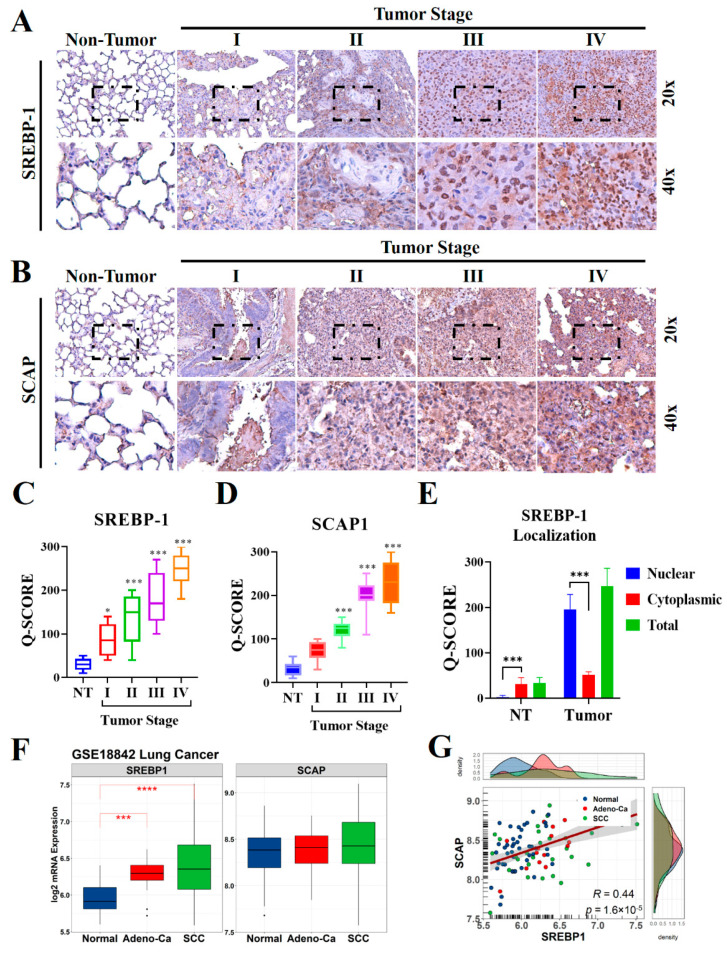
SCAP and SREBP-1 expression in NSCLC and normal lung tissues. Representative IHC images used to measure the differential expression of (**A**) SREBP-1and (**B**) SCAP in NSCLC tissues from patients with stage I, II, III, and IV disease and in non-tumor lung tissue samples. (**C**–**E**) Quantification of SREBP-1 and SCAP expression in non-tumor and tumor lung tissue of different stages, with in-depth analysis of SREBP-1 nuclear localization in lung tissue. (**F**,**G**) Expression and correlation of SREBP-1 and SCAP at the mRNA level in NSCLC samples from the GSE18842 lung cancer microarray dataset compared to the normal lung samples. * *p* < 0.05, *** *p* < 0.001, and **** *p* < 0.0001.

**Figure 2 ijms-23-07283-f002:**
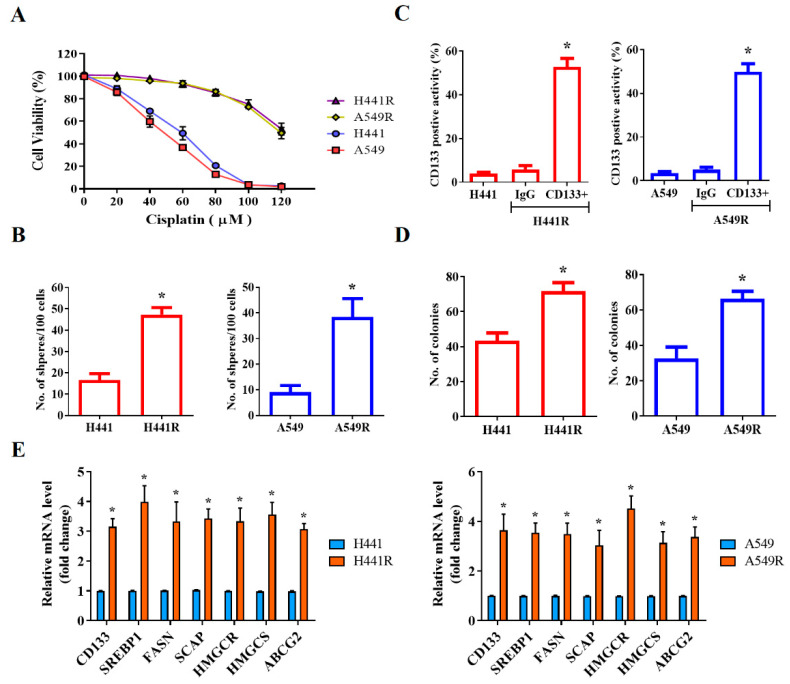
SCAP/SREBP lipogenic signaling axis is associated with cisplatin resistance and cancer stemness in NSCLC cells. (**A**) Comparative MTT assay demonstrating the effect of cisplatin on H441, H441R, A549, and A549R cells. (**B**) Representative flow cytometry images of the levels of CD133 in H441, H441R, A549, and A549R cells. Graphical representation of the number of (**C**) spheres and (**D**) colonies formed by H441, H441R, A549, and A549R cells. (**E**) Histograms indicating the comparative expression of CD133, SREBP-1, FASN, SCAP, HMGCR, HMGCS, and ABCG2 mRNA in H441, H441R, A549, and A549R cells. All data are represented as means ± SDs from assays performed in triplicate at least 3 times. * *p* < 0.05.

**Figure 3 ijms-23-07283-f003:**
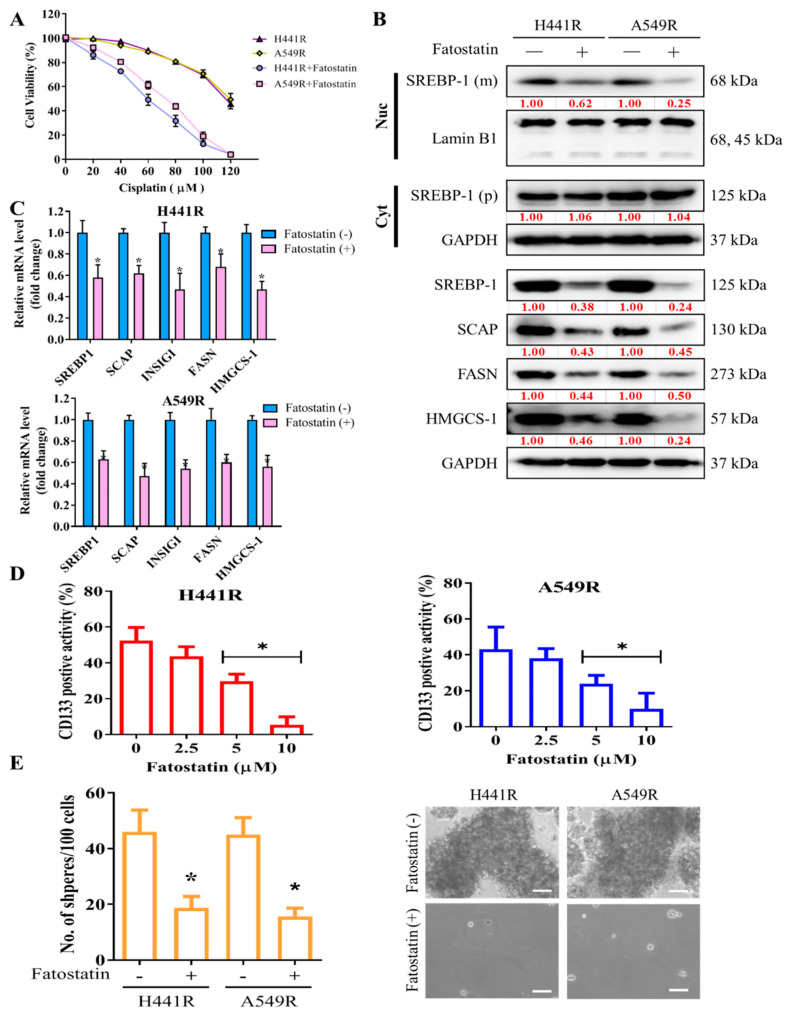
Fatostatin treatment decreased cisplatin resistance and cancer stemness in NSCLC cells. (**A**) Graphical representation of the effect of 20- to 120-μM cisplatin on A549R and H441R cells left untreated or pretreated with 5 µM fatostatin. (**B**) Representative Western blot images indicating protein expression of mature (m) SREBP-1 undergoing cleavage in nuclear (Nuc) compartment (top panel), precursor (p) SREBP-1 in cytoplasm (Cyt), and constitutive expression of SCAP, INSIG1, FASN, and HMGCS-1 in whole lysate of A549R and H441R cells either left untreated or treated with 5-µM fatostatin for 24 h. Lamin B1 and GAPDH were loading control of nuclear and whole lysate, respectively. (**C**) Bar chart showing the comparative expression of SREBP-1, SCAP, FASN and HMGCS-1 mRNA in H441R and A549R cells treatment with/without 5μM Fatostatin. (**D**) Effect of 2.5- to 10-μM fatostatin on the percentage of CD133+ A549R and H441R cells, as demonstrated by flow cytometry. (**E**) Images and graphical representations of the effect of 5-μM fatostatin on the number and size of spheres formed by A549R and H441R cells. Data are represented as means ± SDs from assays performed in triplicate at least 3 times. * *p* < 0.05.

**Figure 4 ijms-23-07283-f004:**
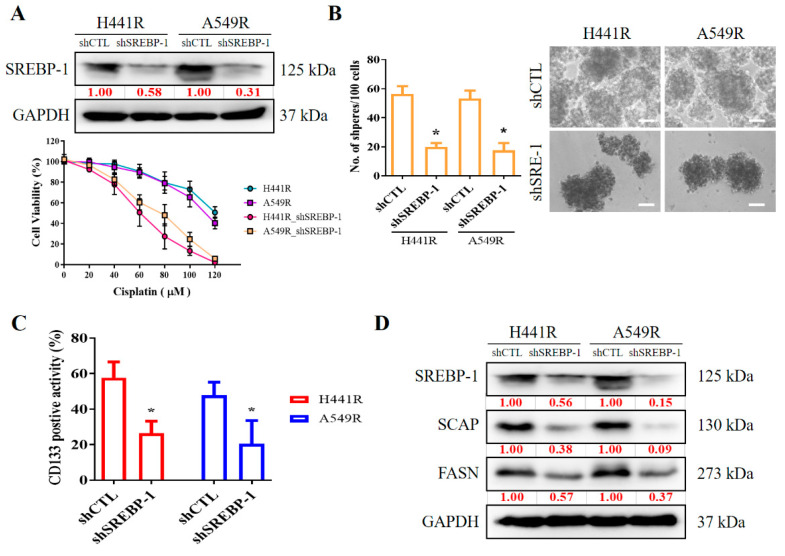
Silencing SREBP-1 attenuated the viability and stemness of H441R and A549R cells. (**A**) Western blot images indicating the knockdown efficacy of shSREBP-1 in H441R and A549R cells. Graphical representation of the effect of cisplatin on A549R, A549R_shSREBP-1, H441R, and H441R_shSREBP-1 cells, as determined through MTT assays. (**B**) Images and graphs of the effect of shSREBP-1 on the number and size of spheres formed by A549R and H441R cells. (**C**) Flow cytometry images indicating how shSREBP-1 affects the percentage of ALDH1+ A549R and H441R cells. (**D**) Western blot analysis images revealing the effect of shSREBP-1 on the expression of SREBP-1, SCAP, FASN, and SREBP2 in transfected A549R and H441R cells relative to their VC counterparts. (**E**) Graphical representation of the effect of shSREBP-1 on the levels of FASN, SCD1, cleaved PARP, and peIF2α in A549R and H441R cells. VC, vector control. Results are represented as means ± SDs from assays performed three times in triplicate. * *p* < 0.05.

**Figure 5 ijms-23-07283-f005:**
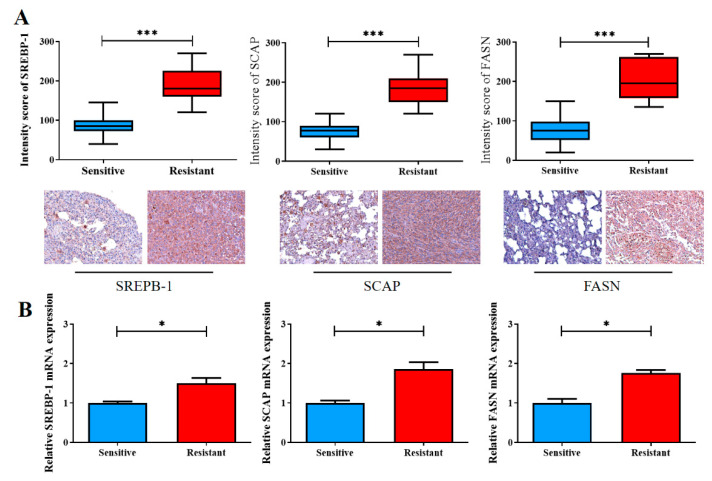
Enhanced SREBP-1/SCAP/FASN signaling is implicated in the reduced sensitivity of NSCLC cells to cisplatin therapy. (**A**) The staining intensity scores of the cisplatin-sensitive and cisplatin-resistant samples were compared. Graphical representations in boxplot and micrographs (with *p* values) indicating the SREBP-1, SCAP, and FASN levels in the cisplatin-sensitive and cisplatin-resistant samples. (**B**) The qRT-PCR analysis was used to measure the expression of SREBP-1/SCAP/FASN. * *p* < 0.05, and *** *p* < 0.001.

**Figure 6 ijms-23-07283-f006:**
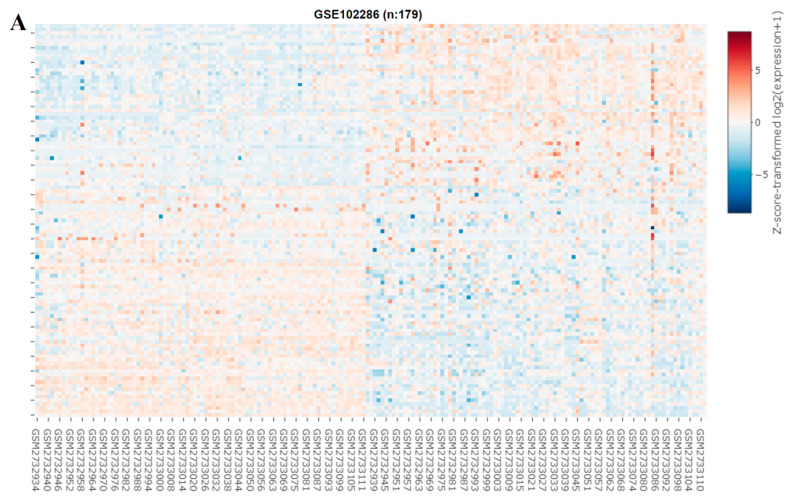
miRNA expression profiles of NSCLC cells. (**A**) Heat map and (**B**) volcano plot of miRNA expression profiles of NSCLC cells. Differentially expressed (DE) miRNAs were identified from the GEO (GSE102286) database containing 179 total lung patient samples using the criteria of |logFC| > 1 and *p* < 0.05. (**C**) Intersection analysis of predicted miRNAs targeting SREBP-1 and SCAP. Predictions were based on the ENCORE database. (**D**) qRT-PCR and (**E**) Western blot images indicating the effect of the introduction of hsa-miR-497-5p (mimic) and its efficacy of hsa-miR-497-5p in targeting SREBP-1 in H441R and A549R cells. * *p* < 0.05.

**Figure 7 ijms-23-07283-f007:**
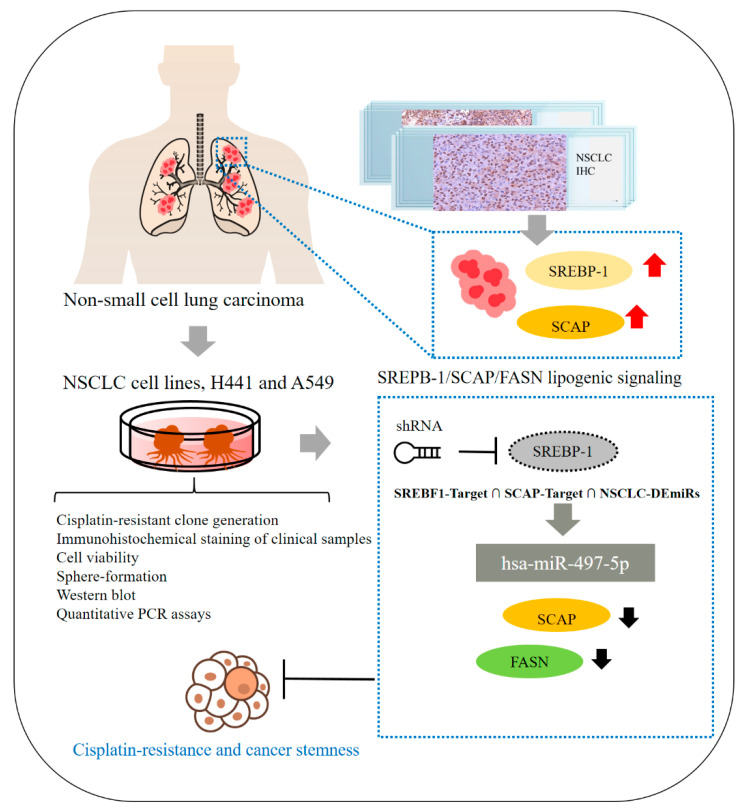
Schematic abstract showing that the role of SREBP-1/hsa-miR-497-5p/SCAP/FASN lipogenic signaling plays a critical role in NSCLC SC-like and chemo-resistant phenotype.

**Table 1 ijms-23-07283-t001:** Association between SREBP-1 and clinicopathological features of NSCLC patients.

Variable (*n* = 98)		SREBP-1 Expression		OR (CI 95%)	*p*-Value
		High	Low		
Gender	Male	27 (47.3%)	30 (52.7%)	1.41 (0.62–3.18) Ref	0.411
	Female	16 (39.0%)	25 (61.0%)		
Age (years)	<50	24 (46.1%)	28 (53.9%)	1.22 (0.55–2.71) Ref	0.629
	≥50	19 (41.3%)	27 (58.7%)		
Smoking history	Yes	17 (45.9%)	20 (54.1%)	2.13 (0.87–5.20) Ref	0.096
	No	14 (28.6%)	35 (71.4%)		
TNM stage	III + IV	29 (59.2%)	20 (40.8%)	**3.63 (1.56–8.41)** Ref	**0.002**
	I + II	14 (28.6%)	35 (71.4%)		
Tumor differentiation	Poor	24 (55.8%)	19 (44.2%)	**2.39 (1.05–5.43)** Ref	**0.035**
	Moderate/well	19 (34.5%)	36 (65.5%)		
SCAP expression	High	22 (56.4%)	17 (43.6%)	**2.34 (1.02–5.36)** Ref	**0.042**
	Low	21 (35.6%)	38 (64.4%)		

OR: odd ratio; Ref: reference group.

## Data Availability

The datasets used and analyzed in the current study are publicly accessible as indicated in the manuscript.

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
