# Peer review of "Targeting the SREBP-1/Hsa-Mir-497/SCAP/FASN Oncometabolic Axis Inhibits the Cancer Stem-like and Chemoresistant Phenotype of Non-Small Cell Lung Carcinoma Cells"

_ijms, 2022, doi:10.3390/ijms23137283_

Round 1
Reviewer 1 Report
In this manuscript, Tiong and colleagues have studied the role of SREBP pathway in maintaining lung cancer stem cell state. The authors showed that SREBP and SCAP are upregulated in lung tumours and reducing their expression makes tumour cells sensitive to chemotherapy. They also identified a microRNA miR-497-5p that targets both SREBP-1 and SCAP in lung tumours.
The authors can improve their manuscript by addressing the following:
1. Most images in the manuscript are blurred and difficult to interpret.
2. In Fig.1, is the localisation pattern of SREBP (cytoplasmic/nuclear) changing in tumours? Also IHC images need to be quantified to properly assess the change in SREBP expression.
3. In Fig. 3, authors show that Fatostatin treatment reduces the levels of SREBP and SCAP. As the drug is mainly known to block post translational changes to SREBP, it is unclear why the absolute levels of these proteins are reduced.
4. In Fig. 5a, does each dot represent a biological replicate? Technical replicates should not be mixed with biological ones.
Author Response
Point-by-point responses to Editor’s comments:
We thank Editor for carefully reading our manuscript and providing valuable comments. We believe making use of all these comments has further helped improve the quality and appeal of our work, as well as strengthened the manuscript. Below are our point-by-point responses.
# General Comments to Editor and all Reviewers
Please kindly find several updates in our respective draft as follows:
- Layout adjustment was done according to current official IJMS layout (Introduction, Result, Discussion, Methods, Conclusion).
- Reference number ordering was automatically changed due to our adjustment according to IJMS layout (Introduction, Result, Discussion, Methods, Conclusion)
- Additional 5 references that previously were unrecognized by our citation manager (all current references in total: 42). Those additional references were mostly cited in the discussion part, such as:
- Sanchez-Palencia, A.; Gomez-Morales, M.; Gomez-Capilla, J.A.; Pedraza, V.; Boyero, L.; Rosell, R.; Fárez-Vidal, M.E. Gene expression profiling reveals novel biomarkers in nonsmall cell lung cancer. International Journal of Cancer 2011, 129, 355-364, doi:https://doi.org/10.1002/ijc.25704.
- Li, A.; Yao, L.; Fang, Y.; Yang, K.; Jiang, W.; Huang, W.; Cai, Z. Specifically blocking the fatty acid synthesis to inhibit the malignant phenotype of bladder cancer. Int J Biol Sci 2019, 15, 1610-1617, doi:10.7150/ijbs.32518.
- Yin, F.; Feng, F.; Wang, L.; Wang, X.; Li, Z.; Cao, Y. SREBP-1 inhibitor Betulin enhances the antitumor effect of Sorafenib on hepatocellular carcinoma via restricting cellular glycolytic activity. Cell Death & Disease 2019, 10, 672, doi:10.1038/s41419-019-1884-7.
- Shen, W.; Xu, T.; Chen, D.; Tan, X. Targeting SREBP1 chemosensitizes colorectal cancer cells to gemcitabine by caspase-7 upregulation. Bioengineered 2019, 10, 459-468, doi:10.1080/21655979.2019.1676485.
- Chen, J.; Wu, Z.; Ding, W.; Xiao, C.; Zhang, Y.; Gao, S.; Gao, Y.; Cai, W. SREBP1 siRNA enhance the docetaxel effect based on a bone-cancer dual-targeting biomimetic nanosystem against bone metastatic castration-resistant prostate cancer. Theranostics 2020, 10, 1619-1632, doi:10.7150/thno.40489.
- Additional methods that were previously not mentioned in the first draft, such as:
- 3. Tumorsphere formation assay
- 4. Activity of CD133 by flow cytometry
- 7. Preparation of nuclear and cytoplasmic lysates
# Comments from Reviewer 1
- Most images in the manuscript are blurred and difficult to interpret.
Answer: We thank the editor for the insightful suggestion. Thereby, we then have already improved the figures provided in a higher resolution and clearer annotation. Please kindly refer to our updated resolution of figure 1 to 6.
- In Fig.1, is the localization pattern of SREBP (cytoplasmic/nuclear) changing in tumours? Also IHC images need to be quantified to properly assess the change in SREBP expression.
Answer: We appreciate the editor for a very positive suggestion. Indeed, the SREBP was highly expressed in tumor than normal, with our analysis also observed an increase of nuclear localization in lung tumor specimens. Therefore, we updated our figure to show the nuclear localization percentage among normal and tumor patients. Then, we have also added other plots to depict our quantification using Q-score to compare level of expression of both SREBP and SCAP between non-tumor and tumor samples. Ultimately, we have re-written the result of subsection 2.1., including figure legend of figure 1, to support the updated result depicted in figure 1. Please kindly refer to our updated subsection 2.1, updated figure 1, and subsequent corrected figure legend.
Updated Figure 1:
Updated Subsection 2.1:
2.1. SREBP-1 and SCAP levels are elevated in patients with NSCLC
To understand the biological significance of SCAP in the development and progression of lung cancer, especially in patients with NSCLC, we evaluated the levels of SCAP and its ligand SREBP-1 in lung cancer specimens (n = 98) through IHC staining (Figures 1A and B). The results indicated that the levels of expression of both SREBP-1 and SCAP in the NSCLC samples were higher than those in the non-tumor samples, and that these elevated levels were stage-dependent (Figures 1C and D).
Updated Figure Legend 1:
Figure 1. SCAP and SREBP-1 expression in NSCLC and normal lung tissues. Representative IHC images used to measure the differential expression of (A) SREBP-1and (B) SCAP in NSCLC tissues from patients with stage I, II, III, and IV disease and in non-tumor lung tissue samples. (C-E) Quantification of SREBP-1 and SCAP expression in non-tumor and tumor lung tissue of different stages, with in-depth analysis of SREBP-1 nuclear localization in lung tissue.
- In Fig. 3, authors show that Fatostatin treatment reduces the levels of SREBP and SCAP. As the drug is mainly known to block post translational changes to SREBP, it is unclear why the absolute levels of these proteins are reduced.
Answer: We appreciate the thoughtful recommendation of the reviewer. Fatostatin interacts directly to the sterol-binding site of SCAP, preventing ER exit of SCAP and ER-to-Golgi transport of SREBP. As a result, fatostatin can decrease the expression of nuclear SREBPs and their downstream genes, which in turn decreases lipid metabolism [1]. Therefore, we could observe a decrease level of protein expression of SCAP in its nuclear mature form SREBP1 following treatment with fatostatin. In this case, in our figure 3, we provided the blot result of mature SREBP1. However, to make it clearer, we reconstructed our figure 3 to distinguish expression level of mature and precursor of SREBP1. Please kindly refer to our updated figure 3 and subsection 2.4.
Updated Subsection 2.4:
Furthermore, the results of western blot analysis showed that treatment with 5 µM Fatostatin reduced expression of cleavage SREBP-1 in the nuclear compartment, along with reduction of SCAP, INSIG1, FASN and HMGCS-1 proteins. On the other hand, the protein expression of cytoplasmic SREBP-1 as precursor form did not change as depicted in the western blot result (Figure 3B).
Updated Figure 3:
Updated Figure Legend 3B:
Figure 3. (B) Representative Western blot images indicating protein expression of mature (m) SREBP-1 undergoing cleavage in nuclear (Nuc) compartment (top panel), precursor (p) SREBP-1 in cytoplasm (Cyt), and constitutive expression of SCAP, INSIG1, FASN, and HMGCS-1 in whole lysate of A549R and H441R cells either left untreated or treated with 5-uM fatostatin for 24 hours. Lamin B1 and GAPDH were loading control of nuclear and whole lysate, respectively.
- In Fig. 5a, does each dot represent a biological replicate? Technical replicates should not be mixed with biological ones.
Answer: We greatly appreciate the attentive suggestion of the reviewer. Indeed, each dot in this figure represented biological replicate of different individual tissue samples. Furthermore, we did not mix our biological replicate with technical replicates. To make it understand easier, we then modified the dot plot into boxplot in order to prevent any misunderstanding in the future. Therefore, please kindly refer to our updated figure 5.
Updated Figure 5:

Reviewer 2 Report
Tung-Yu Tiong et al. showed an interlink between the expression pattern of SREBP1 and its ligand (SCAP) in NSCLC involving biopsies derived from patients is of great value for better understanding the drug resistance and relapsing of these tumors.
The authors also used two different cell lines representing the study model. The authors also used a selective lineage (R) of the cells resistant to Cisplatin or mixed cells (resistance or unresistant) in their study. The study significantly advanced the collective knowledge of SREBP1/SCAP in NSCLC. However, a few points need further clarifications.
Major points:
- Authors must explain the potential reason why fewer patients show tumor differentiation, particularly with high SREBP1 levels and the same for TNM. It seems the levels are inversely related.
- Authors must present the immunoblots from the patient's samples showing SREPB1/SCAP levels to complement the immunostaining pattern presented in figure 1 and offer the quantified result from the patient's samples (sample size is n=50).
- In figure ", the authors used pre-selected Cisplatin resistant clones of A549 or H441 cells. However, the link between stemness and SREBP1/SCAP remained unresolved. Authors must use siSREBP1 or SiSCAP in the assay presented in figure 2, particularly 2B, C and D.
Minor point:
- Title: SREPB1 or SREBP1?
- Line 288, 290, its uM?
- Line 394, SREBP1 instead of SREPB1.
- Rephrase lines 394-396 to make it clearer, particularly what was transfected.
Author Response
# Comments from Reviewer 2
- Authors must explain the potential reason why fewer patients show tumor differentiation, particularly with high SREBP1 levels and the same for TNM. It seems the levels are inversely related.
Answer: We appreciate the editor for a very positive suggestion. To elaborate and deliver our previous result regarding association of SREBP1 expression and tumor differentiation and tumor stage, we then modified Table 1 and calculated the Odd Ratio for each parameter. It resulted that the higher expression of SREBP1 was significantly associated to poor tumor differentiation and advanced cancer stage. Furthermore, we also provided some explanation of those phenomena into our discussion. Please kindly refer to our updated Table 1 and discussion.
Updated Table 1:
Table 1. Association between SREBP-1 and clinicopathological features of NSCLC patients
|
Variable (n = 98) |
SREBP-1 expression |
OR (CI 95%) |
p-value |
||
|
High |
Low |
||||
|
Gender |
Male |
27 (47.3%) |
30 (52.7%) |
1.41 (0.62 – 3.18) Ref |
0.411 |
|
Female |
16 (39.0%) |
25 (61.0%) |
|||
|
Age (years) |
<50 |
24 (46.1%) |
28 (53.9%) |
1.22 (0.55 – 2.71) Ref |
0.629 |
|
≥50 |
19 (41.3%) |
27 (58.7%) |
|||
|
Smoking history |
Yes |
17 (45.9%) |
20 (54.1%) |
2.13 (0.87 – 5.20) Ref |
0.096 |
|
No |
14 (28.6%) |
35 (71.4%) |
|||
|
TNM stage |
III+IV |
29 (59.2%) |
20 (40.8%) |
3.63 (1.56 – 8.41) Ref |
0.002 |
|
I+II |
14 (28.6%) |
35 (71.4%) |
|||
|
Tumor differentiation |
Poor |
24 (55.8%) |
19 (44.2%) |
2.39 (1.05 – 5.43) Ref |
0.035 |
|
Moderate/well |
19 (34.5%) |
36 (65.5%) |
|||
|
SCAP expression |
High |
22 (56.4%) |
17 (43.6%) |
2.34 (1.02 – 5.36) Ref |
0.042 |
|
Low |
21 (35.6%) |
38 (64.4%) |
|||
OR: odd ratio; Ref: reference group
Updated Section 2.2:
The association between SREBP-1 expression and clinical features among patient with NSCLC was described in Table 1. The gender of patient, age, and history of smoking did not associate significantly with expression of SREBP-1 among NSCLC tissue (p > 0.05). Interestingly, higher SREBP-1 expression was significantly associated to poor tumor differentiation (OR 2.39; CI 95% 1.05 – 5.43; p = 0.035) and advanced TNM stage (OR 3.63; CI 95% 1.56 – 8.41; p = 0.002). In addition, we identified significant association of higher SCAP expression and elevation of SREBP-1 level in NSCLC tissue (OR 2.34; CI 95% 1.02 – 5.36; p < 0.042).
Updated Discussion:
The same study identified no direct association between SREBP-1 expression and age or sex [32]. This study demonstrated that SREBP-1 was highly upregulated in poorly differentiated tumor and advanced stage of NSCLC. In line with the previous report, this finding highlights aberrant activation of SREBP-1/SCAP axis to amplify cancer stem cells that are lacking differentiation and depend on lipogenesis, as this axis is also the master regulator of lipid metabolism [32,33]. Poorly differentiated cancers generally increased proliferation rate, thus, frequently found in advanced stage with high incidence of metastasis and are highly invasive.
New Reference:
- Yi, M.; Li, J.; Chen, S.; Cai, J.; Ban, Y.; Peng, Q.; Zhou, Y.; Zeng, Z.; Peng, S.; Li, X.; et al. Emerging role of lipid metabolism alterations in Cancer stem cells. Journal of Experimental & Clinical Cancer Research 2018, 37, 118, doi:10.1186/s13046-018-0784-5.
- Authors must present the immunoblots from the patient's samples showing SREPB1/SCAP levels to complement the immunostaining pattern presented in figure 1 and offer the quantified result from the patient's samples (sample size is n=50).
Answer: We thank for the thoughtful recommendation of the reviewer. According to this suggestion, we added representative immunoblots data in our supplementary figure S1 to confirm consistent finding of the immunohistochemistry data that were previously provided in figure 1. Furthermore, we also added boxplot to depict the quantification of SREBP1 and SCAP according to your suggestion to our updated figure 1 (1C-E). Please kindly refer to our Supplementary Figure S1, figure 1, and sub-section 2.1.
Additional Supplementary Figure S1:
Figure S1. Comparison of SREBP-1 and SCAP expression in tumor and normal counterparts within lung tissue specimens (n=10). (A) Western blotting representative figure from each case of lung cancer specimen denoted increased expression of SREBP-1 and SCAP in tumor than normal site of lung cancer. (B) Bar plot represented elevated expression of both SREBP-1 and SCAP in tumor than the normal parts of 10 cases lung cancer patients. ***p < 0.001, and ****p<0.0001.
Updated Figure 1:
Updated Subsection 2.1:
2.1. SREBP-1 and SCAP levels are elevated in patients with NSCLC
To understand the biological significance of SCAP in the development and pro-gression of lung cancer, especially in patients with NSCLC, we evaluated the levels of SCAP and its ligand SREBP-1 in lung cancer specimens (n = 98) through IHC staining (Figures 1A and B). The results indicated that the levels of expression of both SREBP-1 and SCAP in the NSCLC samples were higher than those in the non-tumor samples, and that these elevated levels were stage-dependent (Figures 1C and D). Similarly, our confirmation through western blotting also observed that SREBP-1 and SCAP were highly expressed in tumor site than non-tumor adjacent site of NSCLC samples (Supplementary Figure S1A and B). Interestingly, predominant nuclear localization was observed in lung tumor samples, which stipulated an intensive nuclear translocation of this transcription factor in lung malignancy (Figure 1E).
- In figure 4, the authors used pre-selected Cisplatin resistant clones of A549 or H441 cells. However, the link between stemness and SREBP1/SCAP remained unresolved. Authors must use siSREBP1 or SiSCAP in the assay presented in figure 2, particularly 2B, C and D.
Answer: We greatly appreciate the attentive suggestion of the reviewer. To address this question, we already added the data to validate knockdown SREBP1 could suppress stemness in both parental and resistant clone of A549 NSCLC cell line according to the reviewer’s suggestion. Please kindly refer to our Supplementary Figure S2 and updated sub-section 2.5.
Additional Supplementary Figure S2:
Figure S2. Loss of SREBP-1 suppresses cancer stemness phenotypes of both parental and resistant clone of A549 NSCLC cell line. (A) Bar plot denoted attenuation of CD133+ activity in shRNA-mediated knock down of SREBP-1 of A549 cell. (B) Bar chart represented suppression of sphere formation following deficiency of SREBP-1 in both parental and resistant A549 cells. (C) Diagram signified reduction of colony forming capacity after inhibition of SREBP-1 of A549 cells, in both parental and resistant clone. *p < 0.05, ***p < 0.001, and ****p<0.0001.
Updated Subsection 2.5:
To confirm the finding, we also compared the perturbation following shRNA-mediated knockdown of SREBP-1 between unsorted parental cell lines and resistant clone that also resulted in an attenuated CD133 activity, reduced generation of tumor sphere, and suppression of colony formation in A549 NSCLC cell line (Supplementary Figures S2A, B, and C). Therefore, these results might indicate to some extent that SREBP-1 determined metabolic axis to provoke stemness phenotype of NSCLC.
- Title: SREPB1 or SREBP1?
Answer: We appreciate the editor for a very positive suggestion. According to most previous study in same research topic, SREBP1 is the most well-known and well-recognized. Therefore, we already corrected all inappropriate annotation into SREBP1 in title and following main text of the manuscript. Please kindly refer to our updated title.
Updated Title:
Targeting the SREBP-1/Hsa-Mir-497/SCAP/FASN Oncometabolic Axis Inhibits the Cancer Stem-Like and Chemoresistant Phenotype of Non–Small Cell Lung Carcinoma Cells
- Line 288, 290, its uM?
Answer: We greatly appreciate the attentive suggestion of the reviewer. Indeed, it should be µM. Therefore, we already made correction to re-write the concentration into µM. Please kindly refer to following line in subsection 2.3.
Revised sentence:
We demonstrated that although the H441 and A549 cells were sensitive to cisplatin treatment, with IC50 concentrations of 58 µM and 47 µM, respectively, H441R and A549R cells were less responsive to the anticancer cytotoxicity of cisplatin, with a concentration of 120 µM producing a cytotoxicity rate of approximately 50% in both cell lines (Figure 2A).
- Line 394, SREBP1 instead of SREPB1.
Answer: We appreciate the editor for a very positive suggestion. Indeed, SREBP1 is the most well-known and well-recognized name for our target. Therefore, we already corrected all inappropriate annotation into SREBP1 in the main text of this manuscript. Please kindly refer to our updated main text with SREBP1 highlighted in red color at multiple spots.
- Rephrase lines 394-396 to make it clearer, particularly what was transfected.
Answer: We thank for the thoughtful recommendation of the reviewer. We already rephrased the sentence into a more understandable diction regarding our transfection protocol. Please kindly refer to our updated sub-section of 4.1 Cell culture and transfection protocol.
Updated subsection 4.1
4.1. Cell culture and transfection
….The H441 and A549 NSCLC cell lines were transfected with short-hairpin RNAs (shRNAs) targeting SREBP-1, specifically at the following sequences: ATCGCTTGCTTCATCGATATT (clone 1), GTGCCTGTTTACCGAACTAAT (clone 2), or GCACCAAATTAGAGAGTCT (clone 3) within DNA sequence of SREBP-1 or a vector (pLKO_TRC005) through Lipofectamine Plus (Invitrogen, Thermo Fisher Scientific) according to the manufacturer’s protocol. The shRNAs were obtained from the National RNAi Core Facility at Academia Sinica Taiwan. Adherent cells were treated with 0.5 mL of the virus, followed by overnight incubation at 37°C in a 5% CO2 atmosphere incubator. On the following day, the viral medium was replaced with fresh medium, and the culture was treated with puromycin (1 µg/mL) to select a population of stably transfected cells.
References
- Shao, W.; Machamer, C.E.; Espenshade, P.J. Fatostatin blocks ER exit of SCAP but inhibits cell growth in a SCAP-independent manner. Journal of Lipid Research 2016, 57, 1564-1573, doi:10.1194/jlr.M069583.

Round 2
Reviewer 1 Report
The authors have addressed my concerns.